# Long-Term Straw Returning Enhances Phosphorus Uptake by *Zea mays* L. through Mediating Microbial Biomass Phosphorus Turnover and Root Functional Traits

**DOI:** 10.3390/plants13172389

**Published:** 2024-08-27

**Authors:** Xiaoyan Tang, Yuxin Zhou, Runjuan Wu, Kuilin Wu, Hui Zhao, Wanyi Wang, Yanyan Zhang, Rong Huang, Yingjie Wu, Bing Li, Changquan Wang

**Affiliations:** 1College of Resources, Sichuan Agricultural University, Chengdu 611130, China; zhouyuxin@stu.sicau.edu.cn (Y.Z.); wurj0205@163.com (R.W.); wukuilin5@163.com (K.W.); yanyan.zhang@sicau.edu.cn (Y.Z.); 14624@sicau.edu.cn (R.H.); yingjiewu@sicau.edu.cn (Y.W.); benglee@163.com (B.L.); 2Linyi Natural Resources Development Service Center, Linyi 276000, China; 3The Agricultural Technology Popularization Station of Chengdu, Chengdu 610041, China

**Keywords:** straw additions, microbial biomass P, P uptake, maize–oilseed rape rotation, root functional traits

## Abstract

The intensive use of chemical fertilizers in China to maintain high crop yields has led to significant environmental degradation and destabilized crop production. Returning straw to soil presents a potential alternative to reduce chemical fertilizer requirements and enhance soil fertility. This study investigates the effects of different nitrogen (N) input levels and straw additions on crop phosphorus (P) uptake and soil P availability based on a long-term N-fertilizer trial. The treatments included no fertilizer input (CK), conventional (NPK), reduced NPK (0.75NPK), and straw-amended (SNPK) treatments. Results indicate that SNPK significantly enhances shoot P uptake and crop yields by 43.7–61.9% and 29.3–39.6%, respectively. The SNPK treatment improved rhizosphere P availability and increased the phosphorus activation coefficient (PAC) by 1.72-fold compared to NPK alone. The enhanced soil P availability under SNPK was primarily attributed to an abundance of functional microbes, leading to higher P storage in the microbial biomass P pool and its turnover. Additionally, SNPK promoted root exudate and phosphate-mobilizing microbes, enhancing P mobilization and uptake. Nitrogen fertilization primarily influenced root functional traits related to P acquisition. These findings provide valuable insights for developing effective fertilizer management strategies in maize–oilseed rape rotation systems, emphasizing the benefits of integrating straw with chemical fertilizers.

## 1. Introduction

The utilization of chemical fertilizers, primarily nitrogen (N) and phosphorus (P), has been a prevalent strategy to sustain crop yields in China over the past few decades, driven by the need to meet the demands of a growing population [1,2]. However, the excessive application of these fertilizers has led to substantial land degradation and severe environmental impacts [3]. The practice of returning straw to fields offers a promising alternative for reducing chemical fertilizer usage and enhancing soil fertility. The nutrients present in straw, including N, P, and potassium (K), can counterbalance the corresponding constituents in chemical fertilizers [4]. 

Straw return has emerged as a significant strategy for reducing N input in the maize–oilseed rape rotation system [5,6,7], influencing soil P cycling [8]. In this context, it is crucial to consider the type of straw being used, as different crops produce straw with varying carbon-to-nitrogen (C/N) ratios. For instance, cereal straws, such as those from maize, have a high C/N ratio, which can slow decomposition unless N is applied to equalize the C/N ratio in the soil [9]. This is particularly relevant for facilitating the decomposition of straw and ensuring that it contributes effectively to soil nutrient cycling.

In this study, we specifically focus on maize and oilseed rape, where the interaction between straw return and N fertilization is critical for optimizing soil P availability and crop yield. However, the impact of the rhizosphere’s plant–soil–microorganism interactions on soil P cycling remains elusive, and P management is often overlooked in the combined strategy of straw return and N input fertilizer management. Therefore, our focus on soil P dynamics under straw return aims to provide valuable insights for effective P management in a maize–oilseed rape rotation system.

Soil microbial biomass phosphorus (MBP) plays a pivotal role in the transformation and biogeochemical cycling of soil phosphorus, as it constitutes a significant source of available phosphorus for plants through the life and death cycles of microorganisms. Microbial biomass turnover is a critical factor influencing P availability [10,11,12]. Straw return is an efficient method of introducing carbon (C) into agricultural systems [13], which stimulates bacterial and fungal growth, thereby enhancing microbial biomass and promoting P release from the microbial pool [10,14]. Spohn and Kuzyakov (2013) found that MBP turnover is primarily driven by the carbon demand of microorganisms [12]. Wang et al. (2016) hypothesized that straw return mediates soil microbial biomass carbon (MBC) and P turnover, enhancing soil P availability under P fertilizer regimes [15]. Their findings indicated that soil organic P levels determined the MBC:MPB ratio, and MBP turnover rates and fluxes were enhanced with straw addition regardless of P input level [16]. While the general effects of NPK and straw fertilization on soil microbial activity are well understood, this study aims to explore the specific impact of these practices on MBP turnover rates and the dynamics of P availability under long-term conditions in a maize–oilseed rape rotation system.

Plants have evolved diverse strategies to cope with P limitation, promoting P acquisition for their growth [17]. These strategies encompass root functional traits such as: (a) root proliferation to augment soil contact volume and increase specific root length (SRL); (b) secretion of acid phosphatase (APase) and organic acids; and (c) enhancement of symbiotic relationships with mycorrhizal fungi [18,19,20]. Straw addition can enhance P uptake by modifying root morphological and physical traits, with the degree of alteration varying among species [21]. How the long-term N-fertilizer regimes (chemical fertilizer vs. straw additions) mediate root P-acquisition strategies and P uptake by crops remains ambiguous.

To explore these questions, we conducted a study during the *Zea mays* L. season based on a 13-year long-term field trial utilizing a maize–oilseed rape rotation system under different N input regimes. Specifically, we examined the following hypotheses: (1) crop yields and P uptake vary under different nitrogen input regimes; (2) the addition of oilseed rape straw to chemical fertilizer can enhance P availability by increasing the MBP pool and activity related to P mobilization; and (3) chemical N fertilization, in combination with oilseed rape straw return, plays a crucial role in governing soil P cycling by controlling root functional traits associated with P acquisition.

## 2. Results

### 2.1. Crop Yield and P Uptake

In 2021, SNPK exhibited the highest P uptake (~40 kg ha^−1^), significantly surpassing other treatments (*p* < 0.01). NPK followed with ~30 kg ha^−1^, while CK and 0.75NPK showed similar lower uptakes (~20 kg ha^−1^). In 2022, SNPK again led (~35 kg ha^−1^), with NPK (~25 kg ha^−1^) outperforming CK and 0.75NPK (~20 and ~15 kg ha^−1^, respectively). The SNPK consistently achieved the highest yields (~11 t ha^−1^) in both years, significantly exceeding other treatments. NPK yielded ~9 t ha^−1^, whereas CK and 0.75NPK yielded ~8 t ha^−1^ (Figure 1).

### 2.2. Soil Nutrient Availability

There were significant effects of fertilizer treatment and soil depth on soil P availability (*p* < 0.01, Figure 2a). Within the 0–20 cm soil depth, Olsen P was significantly higher under SNPK than the other three treatments. For the 0–10 cm soil depth, Olsen P in the SNPK treatment was 1.72-fold and 1.63-fold higher than in the NPK and 0.75NPK treatments, respectively. In the 10–20 cm soil depth, Olsen P was 1.3-fold and 1.42-fold higher under SNPK compared to NPK and 0.75NPK, respectively. In the 20–40 cm soil depth, there were no significant differences in Olsen P across all treatments. There was no significant difference in Olsen P between NPK and 0.75NPK across all fertilizer treatments. Combining the fertilizer treatments, Olsen P in the 0–20 cm depth was significantly greater than in the 20–40 cm depth (*p* < 0.01).

The interaction between fertilizer treatment and soil depth significantly influenced soil NO_3_^−^-N content (*p* < 0.01, Figure 2c). For the 0–30 cm soil depth, soil NO_3_^−^-N was significantly higher in the NPK and SNPK treatments compared to the 0.75NPK treatment. At the 30–40 cm depth, soil NO_3_^−^-N was significantly higher under SNPK compared to NPK and 0.75NPK. Soil NO_3_^−^-N content decreased with increasing soil depth across all treatments.

Similarly, soil NH_4_^+^-N content was significantly influenced by both fertilizer treatment and soil depth (*p* < 0.05, Figure 2b). The SNPK treatment resulted in higher soil NH_4_^+^-N content in the upper soil layers compared to other treatments, and NH_4_^+^-N content decreased with soil depth across all treatments.

Dissolved organic carbon (DOC) was significantly influenced by fertilizer treatment (*p* < 0.05, Figure 2d), but was not significantly affected by soil depth or the interaction between fertilizer and soil depth (*p* = 0.23). DOC levels were higher with straw amendment (SNPK) compared to treatments without straw amendment (NPK, 0.75NPK) across all soil depths.

### 2.3. Soil P Availability

The phosphorus activation coefficient (PAC) was calculated to evaluate soil phosphorus availability across different soil layers and treatments (Table 1). PAC significantly decreased with increasing soil depth within the same treatment. The SNPK treatment had a 3.0-fold and 2.8-fold higher PAC than NPK at 0–10 cm and 20–30 cm depths, respectively. At 10–20 cm, 0.75NPK had a lower PAC than NPK. No significant differences were observed at 30–40 cm among treatments. The CK treatment consistently showed the lowest PAC values, highlighting minimal phosphorus activation without fertilization.

### 2.4. Microbial Biomass and Functionality

The interaction effects between fertilizer treatment and soil depth significantly influenced MBP (Figure 3a, fertilizer × soil depth: *p* < 0.01). There was a significant difference in MBP across all the fertilizer treatments with the highest MBP under straw-amended (SNPK) treatment, regardless of soil depth. The amount of MBP was significantly higher in topsoil (0–10 cm and 10–20 cm) than in subsoil (20–30 cm and 30–40 cm). The amount of microbial biomass C significantly decreased deeper within the soil (Soil depth: *p* < 0.01) and achieved the highest value at the topsoil (0–10 cm) in each of the treatments (Figure 3b). MBC was not significantly affected by the fertilizer treatment (*p* = 0.21). In the topsoil (0–10 cm), the MBC of SNPK was 1.44-fold and 1.25-fold higher than that of NPK and 0.75NPK, respectively. When we compared MBP and MBC between rhizosphere and bulk soil, the amounts of MBP and MBC in the rhizosphere were significantly higher than those in bulk soil (*p* < 0.01), regardless of fertilizer treatment. In the rhizosphere, soil had the highest MBP and MBC content under the straw-amended (SNPK) treatment.

The effects of fertilizer treatment on microbial biomass phosphorus (MBP), total released P, turnover rate, and annual MBP flux are summarized in Table 2. The average MBP was significantly higher in the SNPK treatment (17.6 mg kg^−1^) compared to CK (10.3 mg kg^−1^) and other treatments (*p* = 0.002). Total released P followed a similar trend, with SNPK showing the highest value (8.9 mg kg^−1^, *p* = 0.015). The turnover rate did not significantly differ among treatments (*p* = 0.578). However, the annual MBP flux was significantly higher under SNPK (21.3 kg P ha^−1^ year^−1^) than under CK and other treatments (*p* = 0.013).

The abundance of phosphate solubilizing microorganisms encoding *phoD* and *phoC* genes in the rhizosphere of maize was significantly increased after straw additions (SNPK) (Figure 4g,h). However, long-term NPK input significantly reduced the copies of phosphate-solubilizing functional microbes encoding *phoD* and *pqqC* genes (*p* < 0.01).

### 2.5. Root P-Acquisition Strategies

Fertilizer treatment significantly influenced root exudate traits (Figure 4a,b), root morphological traits (Figure 4c–f) and functional gene copies (Figure 4g,h). The activity of APase followed SNPK > CK > NPK > 0.75NPK (*p* < 0.01; Figure 4a). Straw significantly increased BG activity, which was 2.98-fold and 2.56-fold higher than in NPK and 0.75NPK, compared with NPK (Figure 4b). Straw addition (SNPK) was associated with short thick roots with highest total root length (TRL) and lowest root diameter (RD) (Figure 4c,d), and was accompanied by strong acid phosphatase and β-1,4-glucosidase exudation (Figure 4a,b). However, total root length (TRL), specific root length (SRL), and root tissue density (RTD) under N reduction (0.75NPK) were increased by N reduction (0.75NPK), while roots were characterized by significantly lower APase and BG root exudate quantities (Figure 4a,b) compared to NPK and SNPK.

### 2.6. Drivers of Crop P Uptake under Different Fertilizer Regimes

Partial least-squares path modelling (PLS-SM) revealed that the main drivers for crop P uptake in 2021 varied among different fertilizer treatments (Figure 5). For the CK, NPK, and 0.75NPK treatments, root morphological traits had a significant positive effect on crop P uptake (Figure 5a–c). For SNPK treatment, microbial biomass C and P, phosphate-solubilizing functional microbes encoding *phoD* and *pqqC* genes, and rhizosphere ectoenzyme activities had a significant positive effect on P uptake (Figure 5d).

## 3. Discussion

### 3.1. Response of Shoot P Uptake and Soil P Availability to Different Fertilizer Regimes

Compared to NPK and 0.75NPK treatments, straw additions significantly enhanced shoot P uptake and crop yield of maize (*Zea mays* L.), even though initial soil P availability did not limit crop growth (Olsen P 15.6 mg kg^−1^). This aligns with Wang et al. (2022) [22], who found that straw combined with P addition enhanced yield and P uptake in maize compared to P additions without straw. One rationale is the P release from straw, which can be directly assimilated by plants [23]. The straw added approximately 48.8 kg N ha^−1^, 7.5 kg P ha^−1^, 75 kg K ha ha^−1^, and 3188 kg C ha ha^−1^ to the soil, which likely played a crucial role in enhancing P availability and crop performance. Beyond the nutrients in straw, increased rhizosphere P availability and PAC suggest that straw C inputs boosted the mobilization of stable P and soil bioavailable P [21,24], ensuring ample bioavailable P for maize, a crop particularly sensitive to P deficiency. Crop yield and P uptake were significantly higher in 2021 compared to 2022 under SNPK, while the inverse trend was observed for NPK and 0.75NPK (Figure 1). This suggests that long-term straw additions could lead to stable or even increased crop yields and P uptake.

Straw returning significantly boosted soil P availability in the topsoil (0–10 and 10–20 cm) and rhizosphere P availability compared to chemical fertilizer input alone. Previous studies have shown that plant litter incorporation into soils can enhance labile P content by decreasing the stable P proportion [25,26]. Given that the average P derived from straw in the maize season is 9.2 kg P ha^−1^, and the available and total P contents in the present soil were 60–120 and 1123–1506 kg P ha^−1^, respectively, the contribution of P derived from straw to Olsen P can be considered negligible due to its low content and slow release [15,27]. Furthermore, the higher PAC under SNPK treatment compared to NPK and 0.75NPK indicates that C derived from straw predominantly mediates P cycling, underpinning enhanced P availability with long-term straw additions.

### 3.2. Microbial Biomass and Functional Traits Vary among Different Fertilizer Regimes

In this study, soil MBP was higher with oilseed rape straw additions compared to no straw additions in the topsoil (0–10 cm and 10–20 cm), correlating positively with enhanced MBC content. This can be attributed to microbial growth stimulated by exogenous C derived from straw (3188 kg C ha^−1^), which consequently increased soil MBP through P immobilization by microbes during their growth [28,29]. Exogenous C additions fulfilled the C demand for microbial growth under a C:P ratio lower than 200, alleviating C limitations [30]. Additionally, a positive correlation was observed between microbial biomass and the abundance of *phoD*-/*phoC*-harboring functional microbial groups across all treatments (Figure 5), indicating that a stronger P mobilization capacity tends to provide more P for microbial absorption. The rhizosphere exhibited a higher MBP and MBC than the corresponding bulk soil, with the highest values in the straw addition treatment. The rhizosphere, characterized by a higher amount of C input due to root exudations, consequently, had a higher microbial biomass compared to bulk soil [31,32].

The SNPK treatment resulted in significantly higher total released P (8.9 mg kg^−1^) compared to CK (4.8 mg kg^−1^) and other treatments. This increase in total released P indicates enhanced microbial turnover and P cycling in the soil, supporting higher P availability for plant uptake. Additionally, the annual MBP flux was significantly higher under SNPK (21.3 kg P ha^−1^ year^−1^) than under CK and other treatments, highlighting the role of straw amendments in sustaining higher microbial activity and P release over time.

### 3.3. Root P-Acquisition Strategies in Response to Different Fertilizer Regimes

Fertilizer regimes can shape soil P availability and P uptake by crops by mediating root P-acquisition strategies in terms of root morphology and exudation traits [18,32,33]. Straw additions stimulated higher root exudates activity (APase and BG) (Figure 4a,b,g,h) and enhanced the recruitment of phosphate-mobilizing microbes compared to treatments without straw addition (NPK and 0.75NPK), which could enable P mobilization in the rhizosphere, explaining the higher PAC under SNPK (Table 1). Conversely, roots developed a thinner and longer root system in response to the N-reduction input compared to straw amendment (Figure 4c–f), suggesting P-conservative root traits based on slow elongation under microbial growth [17,34]. This might be due to lower N inputs stimulating root growth, whereas sufficient P content with straw addition depressed root growth in the SNPK treatment. Zhang et al. (2022) [21] found that soil microbial P mediated root P-acquisition strategies depending on crop species. However, under straw addition, maize roots shifted towards P-conservative root traits in this long-term field trial.

### 3.4. Phosphorus Mobilization Drivers Varied among Fertilizer Treatments

Fertilizer regimes determined P uptake and the main drivers of this process. The principal drivers of crop P uptake under chemical fertilizer input were similar. Under NPK treatments, crop P uptake was significantly and positively correlated with enzyme activity and root morphological traits, as revealed by PLS-PM. Under N-reduction treatment, crop P uptake primarily relied on root functional traits, suggesting that roots tend to form low C-cost strategies to acquire nutrients under sufficient P conditions. However, with straw additions, crop P uptake was primarily driven by functional gene copies in the rhizosphere. This indicates that C from straw provides a more favorable environment for microbial growth, resulting in a larger microbial P pool. The P release from MBP turnover could contribute to soil P availability [35,36], especially under sufficient P soil with C limitation for microbial growth and turnover.

## 4. Materials and Methods

### 4.1. Experimental Site

The study was conducted based on the long-term N-fertilization trial in Guanghan County (104.14° E, 31.02° N), Sichuan Province, in Southwestern China in 2009–2022 (Appendix A). The altitude is 323 m. The climate of the site is subtropical monsoon humid, with an average annual temperature of 17.2 °C and an annual precipitation of 1132.1 mm. The soil is classified as a purple soil (termed Regosol in the taxonomy of the Food and Agriculture Organization of the United Nations (FAO) and Entisol in U.S. taxonomy). Soil properties were measured for each experimental plot at a depth of 0–20 cm before sowing in May 2021, and the results are presented in Table 3.

### 4.2. Experimental Design

The field trial was conducted using a split-plot design consisting of four fertilizer regimes randomly arranged in four blocks. The four treatments were as follows: (i) no chemical fertilizer input (CK); (ii) conventional fertilization input level (NPK, i.e., N fertilizer 240 kg ha^−1^, P fertilizer 90 kg ha^−1^, K fertilizer 75 kg ha^−1^); (iii) N-reduction treatment (0.75NPK, i.e., N fertilizer 180 kg ha^−1^, P fertilizer 90 kg ha^−1^, K fertilizer 75 kg ha^−1^); (iv) straw-amended treatment (SNPK, i.e., N fertilizer 180 kg ha^−1^ + whole oilseed rape straw after harvesting, P fertilizer 90 kg ha^−1^, K fertilizer 75 kg ha^−1^). After the oilseed rape harvest, the straw was cut into 3–4 cm small pieces and returned at 7.5 t ha^−1^. Considering the N, P, K, and C content in the oilseed rape straw, this application contributed approximately 48.8 kg N ha^−1^, 7.5 kg P ha^−1^, 75 kg K ha^−1^, 3188 kg C ha^−1^. Nitrogen was supplied to crops as urea (46% N), while P was supplied as Triple Superphosphate (Superphosphate 45). Potassium was supplied as K_2_SO_4_. The fertilization regimes were used every year on *Zea mays* L. for 13 years. The *Zea mays* (Zhongyu No. 3) was sown at the end of May and harvested at the beginning of September in both 2021 and 2022.

### 4.3. Harvest and Measurements

#### 4.3.1. Plant Sampling and Measurements

Fifteen representative maize plants (*Zea mays* L.) were harvested at maturity from each plot in early September of both 2021 and 2022. The samples were air-dried, and the biomass of each plant part was weighed separately. Crop yields of maize were calculated for each plot based on the fifteen representative plants. Shoots were oven-dried at 80 °C for two days and then weighed to determine dry biomass. Subsequently, the shoots were ground, and subsamples were digested in a microwave oven (ETHOS, Milestone) with concentrated HNO_3_ (65%) at 180 °C and 2 MPa. Phosphorus content was determined using the vanado-molybdate method [37].

Root samples from 2021 were washed with tap water and scanned using an EPSON root scanner at 400 dots per inch resolution (Epson Expression 1600 pro, Tokyo, Japan). The total root length (TRL) and average root diameter (RD) were analyzed using Win-RHIZO software (Regent Instruments Inc., Québec, Canada). Specific root length (SRL) was calculated as the ratio of root length to root biomass, and root tissue density (RTD) was obtained by dividing root biomass by root volume.

#### 4.3.2. Soil and Rhizosphere Sampling and Measurements

From May 2021 to August 2022, bulk soil samples were collected six times: in June, September, and December 2021, and March, June, and September 2022. Soil samples were taken from the 0–20 cm depth to investigate microbial biomass phosphorus (MBP) and carbon (MBC).

In August 2021, twelve representative *Zea mays* L. plants from each plot were sampled at maturity, including roots and the surrounding soil. For each plot, rhizosphere soil (1–3 mm soil aggregates adhering to plant roots) and bulk soil (root-free soil) were collected separately. Rhizosphere soil was lightly shaken and brushed off the roots. Bulk soil was sampled from four depths: 0–10 cm, 10–20 cm, 20–30 cm, and 30–40 cm. Each soil sample was divided into three parts for different measurements. The first part was air-dried and ground to determine soil physicochemical properties, including Olsen P, pH, total P (TP), acid phosphatase (APase), and β-1, 4-glucosidase (BG) activities; the second part was stored at −80 °C for the investigation of functional genes related to organic P mineralization (*phoD*, *phoC*, and *pqqC*) using *qPCR*; and the third part was stored at 4 °C for dissolved organic carbon (DOC), NH_4_^+^-N, NO_3_^−^-N concentrations, and microbial biomass measurements.

Soil chemical analyses: The soil samples were extracted with NaHCO_3_ (0.5 M, pH = 8.5) to determine the soil available P across all the soil samples. The P concentrations were measured using the malachite green method as described by Ohno and Zibilske (1991) [38]. The dissolved organic carbon (DOC) was extracted with 0.5 M K_2_SO_4_ (1:5, soil:extractant) following the method of Blair et al. (1995) [39]. Soil nitrate N (NO_3_-N) and ammonium N (NH_4_-N) were determined using field-moist soil and extracted with 1 mol L^−1^ potassium chloride [40] (Keeney and Nelson, 1982). Soil pH was measured by adding deionized water in a 1:5 soil–water ratio, shaking well and standing for 30 min. Soil pH was determined in the water extracts by using a Metrohm-744 pH meter. The soil total phosphorus (STP) content was measured using the molybdenum blue colorimetric method with H_2_SO_4_-HClO_4_ digestion [41] (Murphy and Riley, 1962).

Soil microbial biomass: MBC and MBP were measured using the chloroform fumigation–extraction methods [42,43]. The samples with or without fumigation were extracted with 0.5 M K_2_SO_4_ with a soil:solution ratio of 1:5 for MBP analysis. For MBP, the samples were extracted via the method of Olsen P extraction.

Soil enzyme activities: The activities of APase and BG were determined using p-nitrophenyl phosphoric acid (pPNP) and 4-MUB-β-d-glucoside as substrates following the methods of Neumann and Mori et al. [44,45].

Functional genes: Total soil DNA of the rhizosphere was extracted from 0.25 g of soil using a DNeasy PowerSoil Pro Kit (MO bio Laboratories, California, USA). The extracted DNA was confirmed by agarose gel electrophoresis. The relative abundance of microbes carrying *phoD, phoC*, and *pqqC* genes was quantified using qPCR analysis, and expressed as the copy numbers of the respective genes. Quantitative PCR was carried out with SYBR Premix ExTaqTm Kit (Takara Biotech, Wuhan, China). The primers used and the corresponding amplification protocol for genes are provided in Appendix A.

### 4.4. Statistical Analyses

To characterize the soil P availability and the transformation of P fractions, we calculated the phosphorus activation coefficient (PAC) for each treatment [27]. The PAC was defined as the variations in and the degree of difficulty of the transformations between total P and available P, using the following equation:PAC (%) = AP/TP × 100%
where AP is the content of soil available phosphorus (g kg^−1^) and TP is the content of soil total phosphorus (g kg^−1^).

The MBP turnover rate was calculated by the ratio of the sum of MBP losses and the average MBP [46]. In this study, MBP turnover rate was expressed as per year.
$$MBP\;turnover\;rate=\;\frac{\sum MBP\;losses}{mean\;MBP}$$
where ΣMBP losses is the sum of MBP decreases among two subsequent measurements, and the average MBP is determined as the average MBP for all measurement dates. MBP flux was defined as
MBP flux = mean MBP × ρ ×V×MBP turnover rate
where ρ is the bulk density of the soil (1.25 g cm^−3^) and V is the volume of 1 hectare of the 0–20 cm soil layers [46]. The MBP flux is useful for assessing the amount of P released by microorganisms in a specific period. In this study, the MBP flux was given in kg P ha^−1^ year^−1^ of dry soil.

We firstly tested whether fertilizer regimes (CK, NPK, 0.75NPK, SNPK) and soil depth (0–10 cm, 10–20 cm, 20–30 cm, 30–40 cm) affected soil N, P availability, DOC, and microbial biomass C and P with two-way ANOVA (analyses of variance). Then, differences in root functional traits, enzyme activity, and copies of functional genes between them were tested with one-way ANOVA.

To test the relationship among soil physicochemical properties, enzyme activity, microbial biomass, and functional genes, we constructed a partial least-squares path model (PLS-PM) with the “Vegan” and “PLS-PM” packages in R version 3.6.2 (R Core Team).

## 5. Conclusions

This study demonstrates the benefits of integrating straw amendments with chemical fertilizers (SNPK treatment) in a maize–oilseed rape rotation system. Straw amendments significantly increased shoot P uptake and crop yields, enhancing soil phosphorus availability and microbial biomass phosphorus (MBP). The SNPK treatment showed a higher rhizosphere P availability and phosphorus activation coefficient (PAC), driven by carbon inputs from straw that mobilized stable P forms. Additionally, straw amendments improved microbial activity, total released P, and annual MBP flux, with increased phosphate-solubilizing microorganisms. Root P-acquisition strategies were enhanced by straw amendments, promoting higher root exudate activity and phosphate-mobilizing microbes. These findings underscore the potential of combining organic amendments with chemical fertilizers to improve soil fertility, reduce chemical fertilizer reliance, and promote sustainable crop production. Integrating straw into fertilization practices offers a viable strategy for enhancing P cycling and availability in intensive agricultural systems.

## Figures and Tables

**Figure 1 plants-13-02389-f001:**
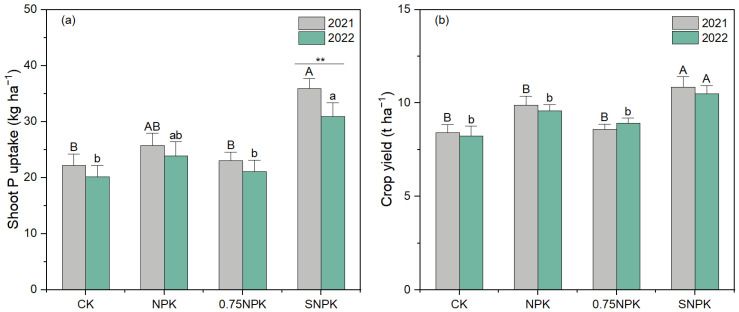
Shoot P uptake (**a**) and crop yield (**b**) of maize in 2021 and 2022 under different fertilizer regimes. Values are means of four replicates (±SE). Two-way ANOVA was performed with fertilizer treatment and year as fixed effects. Different capital letters indicate significant differences (*p* < 0.05) among treatments within 2021. Different lower-case letters indicate significant differences (*p* < 0.05) among treatments within 2022. ** indicates statistical difference at *p* < 0.05.

**Figure 2 plants-13-02389-f002:**
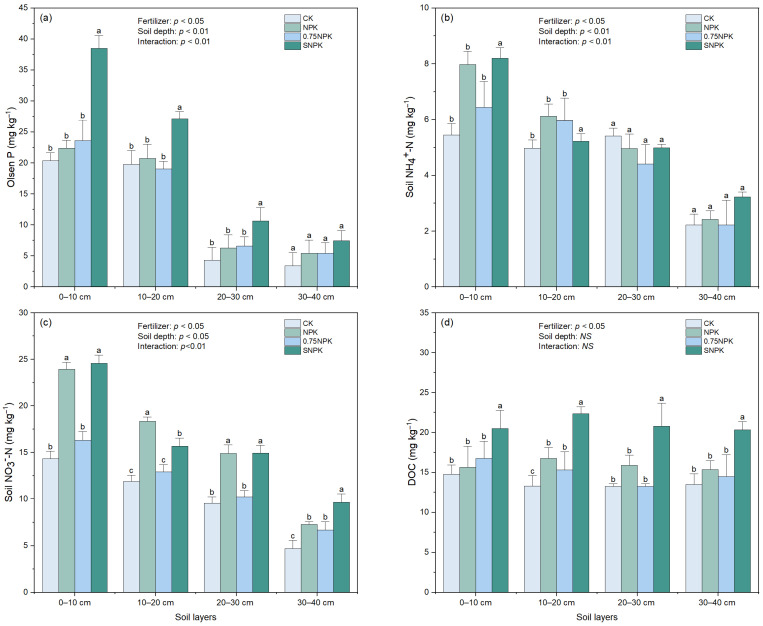
Effect of different fertilizer applications on soil Olsen P (**a**), DOC (**b**), soil NO_3_^−^-N (**c**) and NH_4_^+^-N (**d**) at different soil depths. Two-way ANOVA was performed with fertilization treatment and soil depth as fixed effects. Values are means of four replicates (±SE). Different lower-case letters indicate significant differences (*p* < 0.05) among treatments within the same soil depth. NS means no significant difference.

**Figure 3 plants-13-02389-f003:**
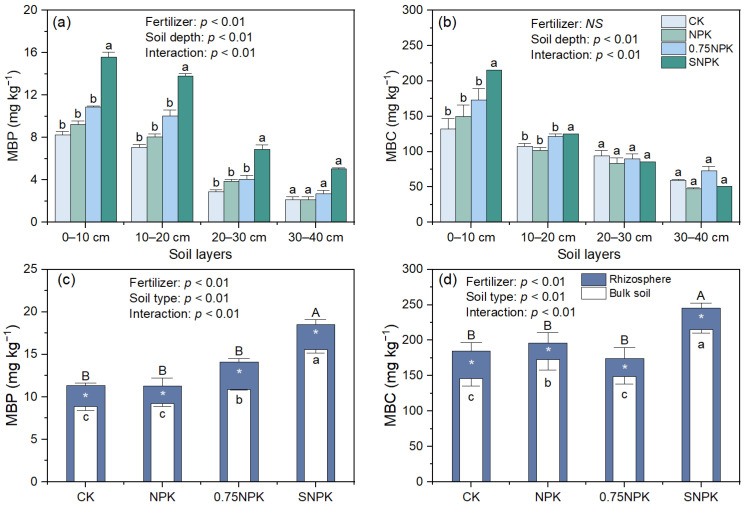
Soil MBP (**a**,**c**) and MBC (**b**,**d**) in different soil depth (**a**,**b**) and in rhizosphere and bulk soil (**c**,**d**). Two-way ANOVA was performed with both fertilization treatment and soil depth as fixed effects (**a**,**b**) and fertilization treatment and soil type (**c**,**d**). Different lower-case letters indicate significant differences (*p* < 0.05) among treatments at the same soil depth (**a**,**b**). Values are means of four replicates (±SE) for MBP, MBC. MBP and MBC for bulk soil are the average values of those at 0–10 and 10–20 cm soil depth (**c**,**d**). Within the N treatment, * denotes a significant difference between rhizosphere and bulk soil (significance: *, *p* < 0.05).

**Figure 4 plants-13-02389-f004:**
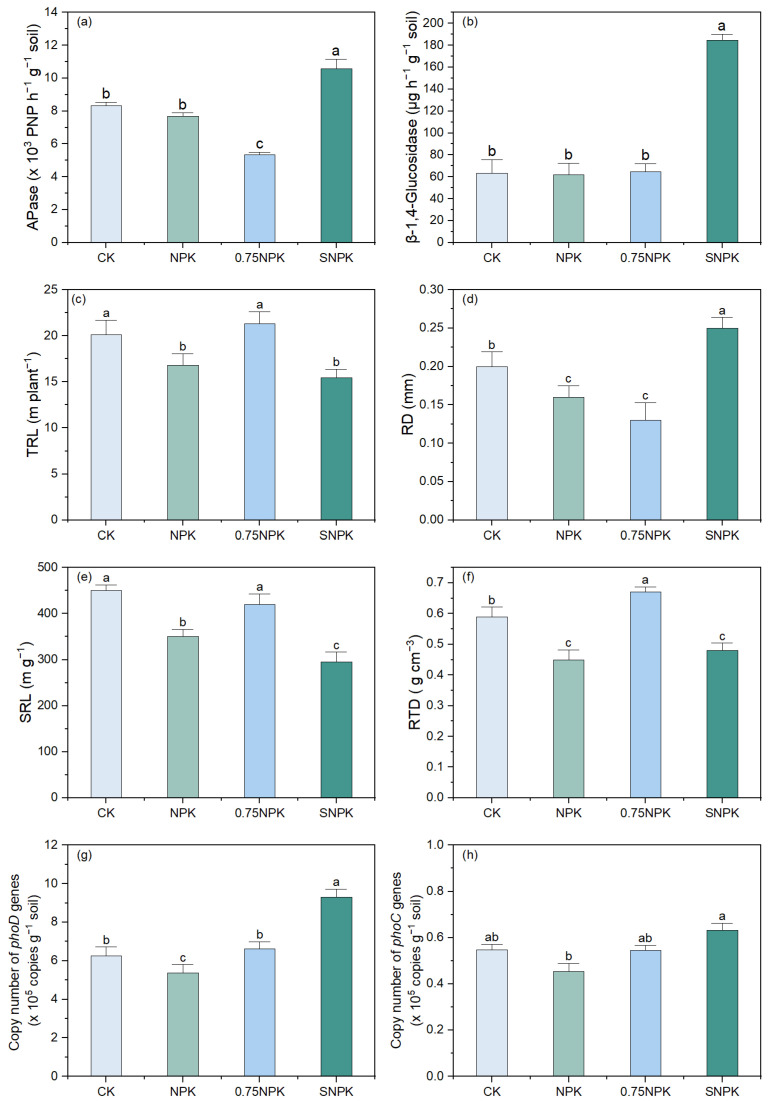
Effects of different fertilizer applications on soil APase (**a**), β-1,4-glucosidase (**b**) in rhizosphere, total root length (TRL) (**c**), root diameter (RD) (**d**), specific root length (SRL) (**e**), root tissue density (RTD) (**f**), copy number of *phoD* (**g**), and *phoC* (**h**) genes in rhizosphere. One-way ANOVA was performed to test the effect of fertilization treatment. Different lower-case letters indicate significant differences (*p* < 0.05) among treatments.

**Figure 5 plants-13-02389-f005:**
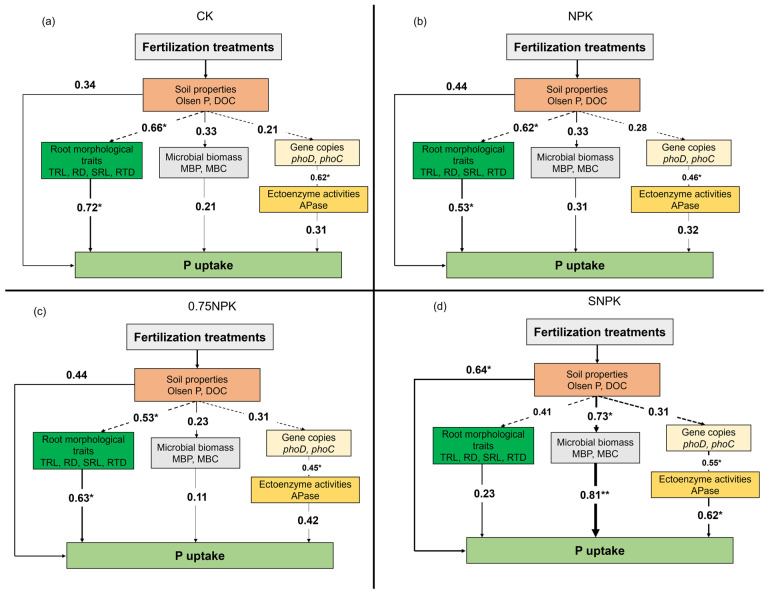
Partial least-squares path modelling (PLS-PM) for crop P uptake under CK (**a**), NPK (**b**), 0.75NPK (**c**), and SNPK (**d**) treatments. The direction of the arrow represents the direction of the path influence, the number on the arrow represents the standardized path coefficient, and the asterisk represents the significance; * indicates the significance level of 0.05; ** indicates the significance level of 0.01.

**Table 1 plants-13-02389-t001:** Phosphorus activation coefficient in different fertilization treatments. Values represent the means of four plots ± SE (standard errors).

Treatment	Phosphorus Activation Coefficient (PAC ^1^, %)			
	0–10 cm	10–20 cm	20–30 cm	30–40 cm
CK	20.4 ± 0.5 b	19.3 ± 0.2 ab	5.16 ± 0.64 b	2.16 ± 0.32 a
NPK	22.6 ± 0.4 b	22.3 ± 0.7 a	4.14 ± 0.51 b	2.14 ± 0.55a
0.75NPK	22.2 ± 0.6 b	16.8 ± 0.3 b	1.75 ± 0.57 b	1.67 ± 0.28 a
SNPK	68.3 ± 0.1 a	21.8 ± 0.1 a	11.7 ± 0.6 a	1.56 ± 0.17 a

^1^ In each column, the phosphorus activation coefficient (PAC) was calculated by the ratio of soil available P (AP) to soil total P content (TP). For a given soil depth, different letters indicate a significant difference between different fertilization treatments using Fisher’s LSD test (*p* < 0.05).

**Table 2 plants-13-02389-t002:** The MBP turnover rates and annual P fluxes through microbial biomass under different fertilization regimes from June 2021 to September 2022.

Treatment	Average MBP (mg kg^−1^)	Total Released P (mg kg^−1^)	Turnover Rate (Year^−1^)	Annual MBP Flux (kg P ha^−1^ year^−1^)
CK	10.3 ± 0.5	4.8 ± 2.6	0.58 ± 0.23	12.0 ± 3.4
NPK	11.3 ± 0.6	5.9 ± 1.0	0.52 ± 0.14	14.7 ± 1.8
0.75NPK	12.5 ± 1.0	6.3 ± 1.1	0.53 ± 0.09	16.3 ± 3.8
SNPK	17.6 ± 0.6	8.9 ± 1.5	0.60 ± 0.1	21.3 ± 4.5
Significance level (*p*)				
Fertilizer	0.002	0.015	0.578	0.013

**Table 3 plants-13-02389-t003:** Selected soil chemical properties of the long-term field trail used in present study. Values represent the means of four plots with SE (standard errors).

Treatments	pH	SOC (g kg^−1^)	TN (g kg^−1^)	AN (mg kg^−1^)	AP (mg kg^−1^)	AK (mg kg^−1^)
CK	5.53 (0.18)	1.30 (0.04)	1.0 (0.11)	99.5 (9.2)	10.5 (1.3)	71.5 (10.2)
NPK	5.54 (0.20)	1.32 (0.02)	1.5 (0.21)	119.8 (13.2)	12.6 (1.5)	73.9 (11.5)
0.75NPK	5.62 (0.04)	1.33 (0.04)	1.3 (0.13)	102.3 (11.2)	13.2 (2.2)	75.1 (8.9)
SNPK	6.02 (0.05)	1.52 (0.02)	1.2 (0.21)	125.2 (33.2)	15.6 (2.1)	79.3 (6.5)

SOC: soil organic matter; TN: total nitrogen; AN: available nitrogen; AP: available phosphorus; AK: available potassium.

## Data Availability

The original contributions presented in the study are included in the article/Appendix A, further inquiries can be directed to the corresponding author/s.

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
