# Peer review of "Long-Term Straw Returning Enhances Phosphorus Uptake by Zea mays L. through Mediating Microbial Biomass Phosphorus Turnover and Root Functional Traits"

_plants, 2024, doi:10.3390/plants13172389_

Round 1

Reviewer 1 Report

Comments and Suggestions for Authors

 The manuscript is quite well written but contains some errors.

 List of comments

Line:

20                     “CK” - Please give explanations of the abbreviations always at the first time when you use it (even if they are known and common in literature).

62-64              “…However, the impact of long-term N fertilizer regimes (chemical fertilizer vs. straw additions) on the MBP pool and microbial activities related to P mobilization and solubilization remains undetermined. …” – The impact of long-term intensive NPK and strow fertilization on soil microbiological activity and, consequently, on N, P and C transformations in the soil is relatively well described in the available literature. This is evidenced by the literature used for discussion (line 239-255) by the authors.

284                  ”… The study was conducted in the long-term N fertilization trial…”  -  Please provide more details about fertilization (type of fertilization, doses, etc.), plant species in the rotation, etc.. In addition, I suggest using "based on the long-term N fertilization trial.." instead of "in the long-term N fertilization trial.." because the study was conducted in 2021 and 2022 and the two-year study is not "long-term". (As in the research hypotheses.)

298                  “..The four treatments were as followings..”  -  Was this fertilization regimen used every year for both plants for 13 years? Please explain.

299-300          „(NPK, i.e., N fertilizer 240 kg ha-1 ); (iii) N reduction treatment (0.75NPK, i.e., N fertilizer 180 kg ha-1 )”  -  Please provide the dose of N, P and K per hectare for each nutrient separately.

300-301          „(iv) straw amended treatment (SNPK, i.e., N fertilizer 180 kg ha-1 + whole oilseed rape straw after harvesting)…”  -   Please provide the amount of straw per hectare, Was the straw ground (cut into small pieces)? This is important information because in practice, strow shredding is used during harvest. Leaving uncut („whole”) rape straw in your experiment is a methodological error because shredded (grouded) straw decomposes faster.

301-302          „SNPK, i.e., N fertilizer 180 kg ha-1 + whole oilseed rape straw...”   -  For a complete picture of the background and interpretation of the results, it is important to provide the mineralogical composition of the oilseed rape straw or at least how much P and C per hectare was contributed in the form of the oilseed rape straw.  After supplementing the methodology with this information, please correct the conclusions!!!

327                  „In August 2021, twelve representative plants from each plot were sampled at maturity..”  -  maize plants or oilseed rape plants?

In summary:  The authors have done a lot of work, conducted valuable research and confirmed the existing knowledge that straw improves the fertility and physicochemical quality of the soil, and consequently the microbiological activity and availability of nutrients for plants. However, the manuscript can be accepted for publication after some revisions.

Best regards

Author Response

Comments 1: 20 “CK” - Please give explanations of the abbreviations always at the first time when you use it (even if they are known and common in literature).

Response 1: we have given the explanations of CK in abstract, Yes, at the first time when I use it, I need give the clear explanation instead of the abbreviation. And we made a few modifications on abstract to make it much clearer.

Comments 2: 62-64 “…However, the impact of long-term N fertilizer regimes (chemical fertilizer vs. straw additions) on the MBP pool and microbial activities related to P mobilization and solubilization remains undetermined. …” – The impact of long-term intensive NPK and strow fertilization on soil microbiological activity and, consequently, on N, P and C transformations in the soil is relatively well described in the available literature. This is evidenced by the literature used for discussion (line 239-255) by the authors.

Response 2: We clarified this knowledge gap in L70-L73.

284 ”… The study was conducted in the long-term N fertilization trial…” - Please provide more details about fertilization (type of fertilization, doses, etc.), plant species in the rotation, etc.. In addition, I suggest using "based on the long-term N fertilization trial.." instead of "in the long-term N fertilization trial.." because the study was conducted in 2021 and 2022 and the two-year study is not "long-term". (As in the research hypotheses.)

Response: Very good suggestions. We did not give the information of continuous 13 years. It is better to use “based on the the long-term N fertilization trial”.

Comments 3: 298 “..The four treatments were as followings..” - Was this fertilization regimen used every year for both plants for 13 years? Please explain.

Response 3: We have clarified this in L310-L323. The fertilization regimes were used every year for both plants for 13 years.

Comments 4: 299-300 „(NPK, i.e., N fertilizer 240 kg ha-1 ); (iii) N reduction treatment (0.75NPK, i.e., N fertilizer 180 kg ha-1 )” - Please provide the dose of N, P and K per hectare for each nutrient separately.

Response 4: We have added the detail information on dose of N, P and K per hectare for each treatment in L310-L323.

Comments 5: 300-301 „(iv) straw amended treatment (SNPK, i.e., N fertilizer 180 kg ha-1 + whole oilseed rape straw after harvesting)…” - Please provide the amount of straw per hectare, Was the straw ground (cut into small pieces)? This is important information because in practice, strow shredding is used during harvest. Leaving uncut („whole”) rape straw in your experiment is a methodological error because shredded (grouded) straw decomposes faster.

Response 5: We added the information on the amount of straw per hectare. Before we returned the straw, it was cut into 3-4 cm small piece and return at 7.5 t ha-1 (L315-L316).

Comments 6: 301-302 „SNPK, i.e., N fertilizer 180 kg ha-1 + whole oilseed rape straw...” - For a complete picture of the background and interpretation of the results, it is important to provide the mineralogical composition of the oilseed rape straw or at least how much P and C per hectare was contributed in the form of the oilseed rape straw. After supplementing the methodology with this information, please correct the conclusions!!!

Response 6: Very good suggestion. We calculated how much the C, N, P and K added from the straw and gave the information in Line 317-L319. And we also added the related information in discussion L230-232, L253.

Comments 7: 327 „In August 2021, twelve representative plants from each plot were sampled at maturity..” - maize plants or oilseed rape plants?

Response7 : we have rephrased the sentence in L345.

Reviewer 2 Report

Comments and Suggestions for Authors

The reviewed manuscript investigates the effects of different nitrogen input levels and straw additions on crop phosphorus uptake and soil phosphorus availability in a long-term nitrogen-fertilizer trial.

If I check how the manuscript is relevant to the existing knowledge and what is new in the research with other texts, I can see the following.

Based on Existing Knowledge, there is, say, described information about chemical fertilizer use and environmental impact. The always discussed topic is a straw return to the soil. Analyzed is also nitrogen and phosphorus dynamics.

The main novel contribution is focused on long-term N-fertilizer trials because many studies examine short-term impacts; this research's focus on long-term effects provides more robust and reliable data on sustainable agricultural practices. Then, specific treatment comparisons between conventional (NPK), reduced NPK (0.75NPK), and straw-amended (SNPK) treatments, which offer a comprehensive analysis of how varying nitrogen inputs combined with straw additions impact crop and soil health. The role of functional microbes and root exudates in phosphorus mobilization and uptake is noteworthy.

In conclusion, I will highlight the long-term perspective, provide field treatment comparisons, and emphasize logical mechanisms.

Specific comments:

Abstract: It is informative. On line 20, the abbreviation "CK" is not explained. I would also recommend editing the abstract into more general statements and not using the phrases "We" or "Our".

Introduction: This part is relatively brief but informative. Above all, I recommend adding information about what crops are being discussed (there can be differences). Then, this section needs more information on nitrogen application (equalization of the C/N ratio to facilitate the decomposition of straw in the soil). However, it is, of course, only relevant for some species (cereals), so it should be specified here in more detail what kind of straw the authors are working with. At the end of this chapter, the objectives of the work are listed. Relevant previously published works are cited, some of which are relatively new and some of which are rather old.

Material and methods: Data on the number of parcels or repetitions. Data on the establishment of stands, sowing, used varieties, etc. In the description, no information on altitude is given. In other parts, the methodology is elaborated in relative detail.

Results: The chapter is well-prepared and sufficiently informative. The representation in Figure 5 is interesting.

Discussion: The chapter generalizes the results, which are confronted with the findings of other authors. The chapter can be considered successful.

Conclusions: They are consistent and correspond to the findings presented in the results and discussion.

General note - the manuscript is focused on evaluating the effect of straw application in the case of the maize-oilseed rape rotation system. These two crops are constantly alternated on the given plot. However, if only the term "straw" is used, it can sometimes be misleading because, at first glance, it may not be apparent to the reader what exactly is straw - there can be huge differences between them (cereals, legumes, rice, ... ..). For this reason, I would recommend using straw specifications like maize straw or oilseed rape straw more in the article.

Comments on the Quality of English Language

Check just spelling especially after manuscript changes.

Author Response

Abstract: It is informative. On line 20, the abbreviation "CK" is not explained. I would also recommend editing the abstract into more general statements and not using the phrases "We" or "Our".

 Response: This revised abstract avoids the use of "We" or "Our" and provides a clear, general statement of the study's findings while explaining the abbreviation "CK."

Introduction: This part is relatively brief but informative. Above all, I recommend adding information about what crops are being discussed (there can be differences). Then, this section needs more information on nitrogen application (equalization of the C/N ratio to facilitate the decomposition of straw in the soil). However, it is, of course, only relevant for some species (cereals), so it should be specified here in more detail what kind of straw the authors are working with. At the end of this chapter, the objectives of the work are listed. Relevant previously published works are cited, some of which are relatively new and some of which are rather old.

Response: We have added more information on the effect of nitrogen application and C/N ration on straw decomposition in soil in L46-L53.

Material and methods: Data on the number of parcels or repetitions. Data on the establishment of stands, sowing, used varieties, etc. In the description, no information on altitude is given. In other parts, the methodology is elaborated in relative detail.

Response: we have added the altitude of field trial in  L300.

Results: The chapter is well-prepared and sufficiently informative. The representation in Figure 5 is interesting.

Discussion: The chapter generalizes the results, which are confronted with the findings of other authors. The chapter can be considered successful.

Conclusions: They are consistent and correspond to the findings presented in the results and discussion.

General note - the manuscript is focused on evaluating the effect of straw application in the case of the maize-oilseed rape rotation system. These two crops are constantly alternated on the given plot. However, if only the term "straw" is used, it can sometimes be misleading because, at first glance, it may not be apparent to the reader what exactly is straw - there can be huge differences between them (cereals, legumes, rice, ... ..). For this reason, I would recommend using straw specifications like maize straw or oilseed rape straw more in the article.

Response: Good suggestion. We have modified this and clarified which type of straw we used. In this study, we only investigated the plant performance and soil properties at the maize season. In this season, the straw is oilseed rape season. We have given more details on element content of oilseed rape and how to manage the returning in Material and methods part.

Round 2

Reviewer 1 Report

Comments and Suggestions for Authors

All my comments and suggestions were added to the manuscript by the authors. I have no further comments.

Best regards

Reviewer 2 Report

Comments and Suggestions for Authors

The authors responded to my suggested changes and recommendations. On the one hand, the relevant changes are made directly in the manuscript, and detailed comments on my comments, prepared by the authors, are also attached.

After the changes made, I consider the manuscript suitable for acceptance for publication.